# A Bulk Oxygen Vacancy Dominating WO_3−x_ Photocatalyst for Carbamazepine Degradation

**DOI:** 10.3390/nano14110923

**Published:** 2024-05-24

**Authors:** Weiqing Guo, Qianhui Wei, Gangrong Li, Feng Wei, Zhuofeng Hu

**Affiliations:** 1GRINM (Guangdong) Institute for Advanced Materials and Technology, Foshan 528000, China; guoweiqing@grinm.com (W.G.); weiqianhui@grinm.com (Q.W.); ligangrong@grinm.com (G.L.); 2Guangdong Provincial Key Laboratory of Environmental Pollution Control and Remediation Technology, School of Environmental Science and Engineering, Sun Yat-sen University, Guangzhou 510006, China

**Keywords:** tungsten trioxide, oxygen vacancy, photocatalytic degradation

## Abstract

Creating oxygen vacancy in tungsten trioxide (WO_3_) has been considered as an effective strategy to improve the photocatalytic performance for degrading organic pollutants. In this study, oxygen vacancies were introduced into WO_3_ by thermal treatment under Ar atmosphere and their proportion was changed by setting different treatment times. WO_3−x_ samples show better photoelectric properties and photocatalytic degradation performance for carbamazepine (CBZ) than an oxygen-vacancy-free sample, and WO_3−x_ with the optimal proportion of oxygen vacancies is obtained by thermal treatment for 3 h in 550 °C. Furthermore, it discovers that the surface oxygen vacancies on WO_3−x_ would be recovered when it is exposed to air, resulting in a bulk oxygen vacancy dominating WO_3−x_ (bulk-WO_3−x_). The bulk-WO_3−x_ exhibited much higher degradation efficiency for CBZ than WO_3−x_ with both surface and bulk oxygen vacancies. The mechanism study shows bulk-WO_3−x_ mainly degrades the CBZ by producing OH radicals and superoxide radicals, while oxygen-vacancy-free sample mainly oxidizes the CBZ by the photoexcited hole, which requires the CBZ to be adsorbed on the surface for degradation. The radical generated by bulk-WO_3−x_ exhibits stronger oxidizing capacity by migrating to the solution for CBZ degradation. In summary, the influence of oxygen vacancy on photocatalytic degradation performance depends on both the proportion and location distribution and could lie in the optimization of the photodegradation mechanism. The results of this study could potentially broaden our understanding of the role of oxygen vacancies and provide optimal directions and methods for oxygen vacancy regulation for photocatalysts.

## 1. Introduction

Along with industrialization and urbanization, organic pollutants have caused severe environmental problems and resulted in increasingly severe shortage of water resources [1]. Therefore, there is urgent demand for developing new techniques for resolving the problem of organic pollutants. In the past decades, research attention has been focused on developing and cost-effective photocatalytic technique as it is a green and efficient approach for degradation of organic pollutants [2,3,4,5,6,7]. In a photocatalytic degradation system, photocatalyst plays a vital role in the degradation efficiency of pollutants. Among all investigated photocatalysts, tungsten trioxide (WO_3_) exhibits a competitive advantage for the degradation of organic pollutants. Owing to its relatively narrow band gap (2.4–2.8 eV) and positive valence band (VB), WO_3_ can utilize visible light energy and have stronger oxidation ability [8,9]. They are both beneficial to oxidize organic pollutants. However, WO_3_, as a photocatalyst, also has the shortcomings of low photogenerated carries density, poor charge transfer efficiency and electron–hole separation [9].

Previous studies have suggested that introducing defect structure into WO_3_ could be an effective way to overcome the above shortcomings, and oxygen vacancy is a simple but effective defect structure for improving the photocatalytic performance of WO_3_ [10]. According to the previous literature, the promotion from oxygen vacancy can be reflected in several aspects. Some research demonstrates that oxygen vacancy improves the utilization of light energy due to extending light absorption of WO_3_ in the visible regions [11]. Some studies demonstrated that oxygen vacancy can effectively promote the separation of photogenerated electron–hole so that it can maximize the utilization rate of photogenerated carriers [12]. Other reports verify that the existence of oxygen vacancy can increase the charge transfer efficiency [13]. However, some reports showed that oxygen vacancies could cause negative influence on the photocatalytic performance because they may act as traps for photogenerated charge carriers [14], become recombination centers of electron and hole and weaken the crystal structure [15].

Oxygen vacancies can be introduced to WO_3_ by thermal treatment in inert or reducing atmosphere [16,17,18,19]. It is investigated whether thermal treatment with different atmospheres, temperatures or durations can result in different impacts on the distribution of oxygen vacancies and, thereby, the photocatalytic performance. When the thermal treatment temperature rises from 350 °C to 550 °C, the formation rate of surface oxygen vacancies is fast at lower temperatures, but it would slow down at higher temperatures because of the migration of oxygen atoms from bulk phase to surface [20]. Higher temperatures may be more conducive to the formation of bulk oxygen vacancies. It is considered that surface and bulk oxygen vacancies generated by thermal treatment play different roles on photocatalytic performance of samples [9]. Bulk oxygen vacancies could format intermediate electronic bands [21,22], which can inhibit the recombination of photogenerated electron–hole pairs [23,24]. Meanwhile, surface oxygen vacancies could result in a decrease in VB due to the local band caused by surface defect [25,26]. In addition, wet chemical reduction, including sol-gel method [27] and solvothermal method [28], and electrochemical reduction [29] are also investigated to construct oxygen vacancies in WO_3_.

This study aims to investigate the influence of oxygen vacancies on the photocatalytic degradation performance of WO_3_. Here, carbamazepine (CBZ) will be used as a representative pollutant for studying their photocatalytic degradation performance. The proportion of introduced oxygen vacancies into WO_3_ are changed by setting different thermal treatment times under Ar atmosphere and optimized by analyzing their structures and morphology properties, photoelectrochemistry properties and photocatalytic degradation performance for CBZ. Furthermore, the influence of oxygen vacancy location is also considered by comparing the WO_3−x_ samples with different distributions of surface and bulk oxygen vacancy. Finally, the CBZ degradation mechanism by WO_3_ and optimized WO_3−x_ are investigated.

## 2. Materials and Methods

### 2.1. Preparation of Pristine WO_3_ and WO_3_ with Oxygen Vacancies

A pristine WO_3_ sample was prepared by a hydrothermal method, referring to reported reference [30]. First, 210 mg of sodium tungsten (Na_2_WO_4_, Sigma-Aldrich, Burlington, MA, USA) was dissolved in 30 mL of deionized (DI) water. Then, 10 M hydrochloric acid (HCl, Fischer Scientific, Waltham, MA, USA) was added dropwise into the solution until it became turbid, followed by the addition of 210 mg of ammonium oxalate ((NH_4_)_2_C_2_O_4_, Sigma-Aldrich, Burlington, MA, USA) into the mixture. After 10 min of stirring, the turbid solution became clear and was then transferred into a Teflon-lined stainless-steel autoclave, which was sealed and heated at 120 °C for 12 h. After that, the resulting yellow powder was filtered and washed with ethanol and water. The yellow powder was WO_3_∙H_2_O and transformed into WO_3_ after being calcinated at 550 °C for 3 h.

WO_3−x_ samples: the pristine WO_3_ nanoparticles were thermally treated in pure Ar flow at 550 °C for 0.5 h, 3 h, 6 h or 10 h to prepare four WO_3−x_ samples, which were denoted as WO_3−x_-0.5 h, WO_3−x_-3 h, WO_3−x_-6 h or WO_3−x_-10 h samples, respectively.

### 2.2. Preparation of Pristine WO_3_ and WO_3_ with Oxygen Vacancies

Scanning electron microscopy (SEM) images were taken on a Quanta 400 Thermal FE environmental scanning electron microscope (FEI, Eindhoven, The Netherlands). The diffuse reflectance absorption spectra of the samples were obtained by a UV-visible spectrophotometer equipped with an integrated sphere attachment (UV-3600, SHIMADZU, Kyoto, Japan). The X-ray diffraction (XRD) data were recorded on a Rigaku diffraction instrument using a Cu Kαradiation source with λ = 0.15418 nm (SmartLab, Rigaku, Tokyo, Japan). Photoelectrochemistry (PEC) measurements were performed on an electrochemical workstation (CHI760E, Chenhua, Shanghai, China) equipped with a three-electrode cell. The working electrode was made as follows: 2 mg of samples was fixed on FTO conducting glass mixed with 0.05 mL 0.05 wt% Nafion (Sigma-Aldrich, Gillingham, Dorset, UK), using graphite rod as counter electrodes and Ag/AgCl as the reference electrodes, and 0.1 M Na_2_SO_4_ was used as the electrolyte. The EPR spectra were detected by a CW/Pulse EPR system (A300, Bruker Co., Bremen, Germany) with a microwave frequency of 9.64 GHz, a microwave power of 0.94 mW, a modulation frequency of 100 kHz and a modulation amplitude of 2.0 G. X-Ray photoelectron spectroscopy (XPS) was performed using a Thermo Fisher Scientific K-Alpha (Thermo Scientific, Waltham, MA, USA) equipped with a VG CLAM 4 MCD electron energy analyzer (Thermo Scientific, Waltham, MA, USA) and twin-anode Mg Kα radiation (1253.6 eV) or Al Kα radiation (1496.3 eV) X-ray sources.

### 2.3. Photocatalytic Performance Experiments

All photocatalytic degradation experiments were carried out in a quartz tube, with 0.25 g/L photocatalyst and 5 ppm carbamazepine (CBZ, Macklin, Shanghai, China) under ultraviolet-visible irradiation from a 35 W xenon lamp. The control measurement of CBZ under illumination was tested as the blank sample. At an interval of 20 min, 0.5 mL of the solution was extracted. The residual concentrations of CBZ in reaction solution were detected every 20 min by HPLC (Shimadzu LC-20AD, Shimadzu Corporation, Kyoto, Japan) equipped with a Poroshell 120 EC-C18 column (4.6 mm × 100 mm, 2.7 μm, Agilent Technology, Santa Clara, CA, USA). The mobile phase for HPLC detection was a mixture of methanol and water with a ratio of 6:4 and at a 0.6 mL/min flow. The photocatalytic activity was evaluated using a time profile of C_t_/C_0_, where C_t_ is the concentration of the pollutants at the irradiation time t and C_0_ is the concentration at the equilibrium before irradiation, respectively.

### 2.4. Detection and Measurements of Photogenerated Radicals

The formed active oxygen radicals, including ·OH, ·O^2−^ and ^1^O_2_, in photocatalytic degradation systems were all identified by a Bruker A300 electron paramagnetic resonance (EPR) spectrometer (Billerica, MA, USA) using 5,5-dimethyl-1-pyrroline-N-oxide (DMPO), DMPO with methanol and TEMP as the trapping agent. Typically, 100 mM of DMPO was mixed with 20 mL solution (including 0.25 g∙L^−1^ catalyst, 5 ppm CBZ). All samples had been illuminated for 5 min before ERP measurements.

## 3. Results and Discussion

### 3.1. Structures and Morphologies Properties

The SEM images show that the WO_3_ powder particle is a cluster of two-dimensional nanosheets with the size of ca. 1 μm (Figure 1a,b). After being thermally treated for 0.5 h to 10 h, there is no remarkable visual change in particle surface morphology and size of the samples (Figure 1c–f), suggesting the morphology of WO_3_ is stable.

Figure 2 shows the optical properties of all samples. As compared with untreated WO_3_, the color of thermally treated WO_3_ samples changed from yellow to olive-green (0.5 h and 3 h) and then to indigo after 6 h and 10 h thermal treatment (Figure 2a). According to the literature, the change in color could be attributed due to the formation of bulk oxygen vacancies [9]. Correspondingly, their light absorption in the visible regions is extended with the increase in oxygen vacancy (Figure 2b). It can be found that more light can be absorbed when the WO_3_ samples are thermally treated for a longer time. Figure 2c shows the band gap of the samples calculated by the linear approximation of (αhν) 2 vs. photo energy [31]. Visibly, all WO_3−x_ samples possess a narrower band gap than pristine WO_3_ due to oxygen vacancy [9,22,32]. The calculated value of pristine WO_3_ is around 2.77 eV, whereas the optical absorption of the WO_3−x_ samples gradually shifts to the longer wavelengths with the band gap value (*Eg*) ranging from 2.53 to 2.71 eV. Among all WO_3−x_ samples, WO_3−x_-10 h shows the most intense light absorption and the narrowest band gap.

The XRD pattern (Figure 3a) of all samples could be indexed to the monoclinic phase of WO_3_ (JCPDF20-1324). There is no observable shift for all characteristic diffraction peaks of WO_3_ after thermal treatment. It suggests that the main crystal structure is preserved even though oxygen vacancy is increased [33]. Nevertheless, the intensity of the diffraction peaks slightly weakens with the increase in their thermal-treated time, indicating the degradation of the WO_3_ crystallinity [34]. Crystalline degradation was mainly reflected in the orientations on (020), (022) and (202) planes (Figure 3b,c).

Based on the results of SEM, UV-vis diffuse reflectance spectra and XRD, it can be concluded that the introduction of oxygen vacancy has little influence on the surface morphology and size of WO_3_, except narrowing its band gap and weakening its crystal structures.

### 3.2. Photoelectrochemistry (PEC) Properties

PEC measurements were conducted in a three-electrode electrochemical cell in 0.1 M Na_2_SO_4_ solution (pH: ca. 7.0). The photocurrent density is measured at 0.6 V and the result is shown in Figure 4a. It illustrates that WO_3−x_ samples can produce higher photocurrent density than pristine WO_3_, as reflected in the i-t patterns with on–off recycles of intermittent irradiation. This suggests that the introduction of oxygen vacancy, to some extent, can improve the photo-irradiated charge transfer efficiency of WO_3_. The improvement is considered to come from WO_3_/WO_3−x_ heterojunction formed by the energy difference between WO_3_ and the oxygen-vacancy-rich layer [35]. Among all WO_3−x_ samples, WO_3−x_-3 h exhibits the highest photocurrent density, which is about triple that of pristine WO_3_ at 0.6 V vs. RHE. Nevertheless, the photocurrent density of WO_3−x_ samples decreases when the oxygen vacancies exceed the optimal proportion, indicating that the further increase in oxygen vacancies in WO_3−x_ is not necessary.

A similar trend has been observed in EIS, which is the common analytical tool to estimate charge separation and migration. As displayed in Figure 4b, four WO_3−x_ samples exhibit a smaller arc radius than pristine WO_3_, verifying that the existence of oxygen vacancy can reduce interfacial charge transport resistance, thus leading to more effective charge separation. The charge transport resistance reached the minimum value when WO_3_ had been thermally treated for 3 h. A trend from rising then declining generally means the charge transfer efficiency of WO_3−x_ samples suffers from two opposite (positive and negative) influence factors. Based upon the result of XRD (Figure 3), the negative influence may arise from the crystalline destruction when the samples have excessive oxygen vacancies. As a result, the optimal proportion of oxygen vacancies appeared when WO_3_ was thermal treated for 3 h and, thus, WO_3−x_-3 h presents the best electrical performance.

The Mott–Schottky plots (Figure 4c) of pristine and WO_3−x_ samples are plotted according to ref [30,36,37]. The flat-band potentials are calculated to be 0.03, 0.09, 0.13, 0.07 and 0.06 V vs. NHE for WO_3_, WO_3−x_-0.5 h, WO_3−x_-3 h, WO_3−x_-6 h and WO_3−x_-10 h, respectively. As their *Eg* were 2.77, 2.71, 2.67, 2.62 and 2.53 eV, their VB can be calculated according to the equation of *E*_g_ = *E*_VB_ − *E*_CB_. The detail of band structures is illustrated in Figure 4d.

### 3.3. Photocatalytic Degradation of CBZ

The photocatalytic degradation of 5 ppm CBZ by as-prepared samples is investigated, and results are shown in Figure 5 and Figure 6 and photocatalytic degradation efficiency (%) can be calculated by Equation (1) [38]. Here, C_t_ is the concentration of the pollutants at the irradiation time t and C_0_ is the concentration at the equilibrium before irradiation, respectively.
Degradation efficiency (%) = (1 − C_t_/C_0_) × 100%(1)

As showed in Figure 5, only 36% CBZ has been removed by pristine WO_3_ after 100 min, while those of the WO_3−x_ samples ranged from 49% to 94%. Obviously, the degradation efficiencies on WO_3−x_ samples are remarkably higher than that of pristine WO_3_. Here, the kinetic constant (*k*) for CBZ is further estimated by the following equation [38]:*k* = −ln(C_t_/C_0_)/t(2)

The plots of −ln(*C*_t_/*C*_0_) vs. t are found to be a linear relationship, clarifying that the photocatalytic degradation reaction of CBZ can be described by using a pseudo-first-order model (Appendix A). From the kinetic curves, the kinetic constant k values on WO_3_ are calculated to 0.0060 min^−1^, while those of WO_3−x_-0.5 h, WO_3−x_-3 h, WO_3−x_-6 h and WO_3−x_-10 h are increased to 0.0106, 0.0340, 0.0212 and 0.0081 min^−1^, respectively (Figure 6a). Hence, the k value on WO_3−x_-3 h was the largest and 5.67 times that of on WO_3_, suggesting that the introduction of oxygen vacancy can significantly improve the photocatalytic degradation performance of WO_3_. However, the photocatalytic performance of WO_3−x_ samples decreased when the oxygen vacancies exceeded the optimal proportion (WO_3−x_-3 h). The improved photocatalytic performance is considered to originate from the promotion of the dissociation of excitons and accelerates the transfer of charges.

### 3.4. Optimization of Oxygen Vacancy

The activity of WO_3−x_ can also be enhanced by optimizing the oxygen vacancy location distribution. As mentioned above, the thermal treatment causes oxygen vacancies both in bulk and on surface and leads to higher activity. Interestingly, it was found that the sample would exhibit even higher activity when it was placed in the air for a period of time. This interesting phenomenon can presumably be attributed to the recovery of the surface oxygen vacancies. It may produce bulk oxygen vacancy dominating WO_3−x_ samples, along with a decrease in surface oxygen vacancies, and, hence, these were named as bulk-WO_3−x_. The degradation efficiency of bulk-WO_3−x_ is higher and their kinetic constant increases to 0.0271, 0.0628, 0.0284 and 0.0208 min^−1^. They are 1.34~2.57 times that of WO_3−x_ just after thermal treatment. This indicates oxygen exposure of the thermally treated sample will exhibit higher activity as compared.

The surface oxygen vacancies can be investigated by XPS. Figure 6b–d show the O 1s XPS spectra of WO_3_, WO_3−x_ (represented by WO_3−x_-3 h) and bulk-WO_3−x_ (represented by bulk-WO_3−x_-3 h) samples, which were deconvoluted in three peaks. The peaks at about 530.2, 531.7 and 533.2 eV correspond to surface lattice oxygen species (O_L_), hydroxyl species (O_OH_) and adsorbed water species (O_H2O_), respectively. It is considered that the percentage of O_OH_ signal links to the quantity of surface oxygen vacancies [9,24,39]. The percentages of the three oxygen species of WO_3_, WO_3−x_ and bulk-WO_3−x_ samples are listed in Table 1. Their percentage of O_OH_ increases obviously from 11.8% for WO_3_ to 15.3% for bulk-WO_3−x_, suggesting the thermal treatment causes surface oxygen vacancies.

Moreover, when the WO_3−x_ sample was placed in the air for a period of time, the density of surface oxygen vacancies decreases apparently. As shown in Table 1, the percentage of O_OH_ decreases from 15.3% to 11.3%. This demonstrates that many surface oxygen vacancies are recovered. It is reported that the surface oxygen vacancies will cause a shift of VB position [9,25,26]. However, the VB positions of WO_3_ and bulk-WO_3−x_ samples are both 2.80 eV (Figure 6f), without any shift in VB positions between the two samples. It demonstrates that much of the surface oxygen vacancies are recovered and the density of surface oxygen vacancies in bulk-WO_3−x_ sample has become similar to that of pristine WO_3_. Meanwhile, EPR spectra indicate the formation of bulk oxygen vacancies (Figure 6e). The EPR signal can still be observed on bulk-WO_3−x_ sample but there is no signal in the pristine WO_3_ sample. It indicates the existence of bulk oxygen vacancy in the bulk-WO_3−x_ sample.

Therefore, it can be concluded that the activity of samples increases when both bulk and surface oxygen vacancies are presented, while the activity is further increased when the surface oxygen vacancies are recovered. The procedures of preparation from pristine WO3 to WO_3−x_ and bulk-WO_3−x_. are exhibited as Figure 7.

### 3.5. Mechanism of Photocatalytic Degradation

Inspired by this interesting phenomenon, the bulk oxygen vacancy dominating samples is considered as the optimal one for photocatalytic degradation. Subsequently, the mechanism for the degradation is studied.

To investigate the contribution of the various oxidative species generated in the photocatalytic system of WO_3_ and bulk-WO_3−x_ samples for the degradation of CBZ, active species are tested by EPR. The generation of reactive oxygen species in the system are monitored though EPR technique after adding DMPO or TEMP as the trapping agent under UV-vis radiation. As shown in Figure 8a, a four-line EPR signal of DMPO-·OH adduct with an intensity ratio of 1:2:2:1 is also detected in both of WO_3_ and bulk-WO_3−x_ systems, and the intensity of DMPO-·OH adduct signal in the system with WO_3−x_ is much stronger than that of WO_3_. It indicates oxygen vacancy is significantly conducive to the generation of ·OH. As the VB of WO_3_ and WO_3−x_ are both 2.80 eV, more positive than the standard redox potential of H_2_O/·OH (2.37 V) [40], both samples can oxidize H_2_O to ·OH under UV-vis thermodynamically. Therefore, ·OH can be generated from the oxidation of H_2_O directly (h^+^ + H_2_O → ·OH + H^+^). Then, methanol is added during the EPR measurement to catch ·OH and four very weak peaks with an intensity ratio of 1:1:1:1 are detected in bulk-WO_3−x_ with oxygen vacancies (Figure 8b). It suggests that ·O^2−^ is generated in bulk-WO_3−x_ systems, as they are the characteristic quartet signal of DMPO-·O^2−^ adduct. Here, ·O^2−^ may be generated by the reduction of O_2_ (e^−^ + O_2_ → ·O^2−^). However, these characteristic signals cannot be observed in WO_3_ systems without oxygen vacancies. Therefore, this strongly indicates that the introduction of oxygen vacancies is favorable for the formation of ·O^2−^, which is beneficial to the degradation efficiency. In addition, there is no signal of TEMP-^1^O_2_ adduct detected in WO_3_ nor bulk-WO_3−x_ systems (Figure 8c).

In addition, active species trapping tests are also executed by adding KI and K_2_Cr_2_O_7_ as the quenchers of photoexcited hole (h^+^) and e^−^, respectively. When 10 mM KI is added into the WO_3_ system, the degradation efficiency of CBZ remarkably decreases to 66.2% of pristine value (Figure 8d). It suggests that photoexcited hole (h^+^) is one of the major reactive species contributing to the catalytic degradation of CBZ by WO_3_. However, the addition of KI in the bulk-WO_3−x_ does not influence the degradation efficiency. This means that the hole does not directly cause the degradation of CBZ; instead, the hole mainly oxidized water into OH radicals for the degradation.

Meanwhile, the degradation efficiency of CBZ remarkably increase to 197.6% as compared with pristine value when 10 mM K_2_Cr_2_O_7_ is added. The increase may originate from the promotion of charge separation due to quenching of e^−^. In comparison, the degradation efficiency of CBZ in the bulk-WO_3−x_ photocatalytic system has little change by adding 10 mM K_2_Cr_2_O_7_. This should be due to the high charge transfer efficiency of WO_3−x_ resulting in insignificant enhancement caused by the introduction of K_2_Cr_2_O_7_.

Based on the above analysis, the bulk oxygen vacancy has optimized the photodegradation mechanism of CBZ by WO_3_. The oxygen-vacancy-free sample mainly oxidizes the CBZ by the photoexcited hole (h^+^) (Figure 9a), requiring the CBZ to be adsorbed and cause low activity. By contrast, the bulk oxygen vacancy dominating WO_3_ mainly produces ·OH or ·O^2−^ to degrade CBZ; the stronger oxidization power of radicals and their ability to migrate to the solution is beneficial to the degradation of CBZ (Figure 9b).

## 4. Conclusions

In summary, introduction of oxygen vacancy can improve photoelectric properties and photocatalytic degradation performance of WO_3_ samples. The photocurrent density and degradation efficiency of WO_3−x_ increase firstly, followed by diminishing, along with an increase in the proportion of oxygen vacancy. The bulk and surface oxygen vacancies are produced in WO_3−x_ just after thermal treatment. It is found that surface oxygen vacancies would be recovered when WO_3−x_ was oxygen-exposed, resulting in bulk oxygen vacancies dominating WO_3−x_, whose photocatalytic degradation performance even shows further improvement. Therefore, both proportion and location distribution of oxygen vacancy could exert different influences on the photocatalytic performance of WO_3−x_. The recovered behavior of surface oxygen vacancy and more essential influence of bulk oxygen vacancy deserve to be further studied. The CBZ degradation mechanism was compared between WO_3_ and bulk oxygen vacancies dominating WO_3−x_. The results suggest that improvement in photocatalytic degradation may be due to the generation of OH radicals and superoxide radicals. The results of this study could potentially broaden our understanding of the role of oxygen vacancies and provide optimal directions and methods for oxygen vacancy regulation for photocatalysts.

## Figures and Tables

**Figure 1 nanomaterials-14-00923-f001:**
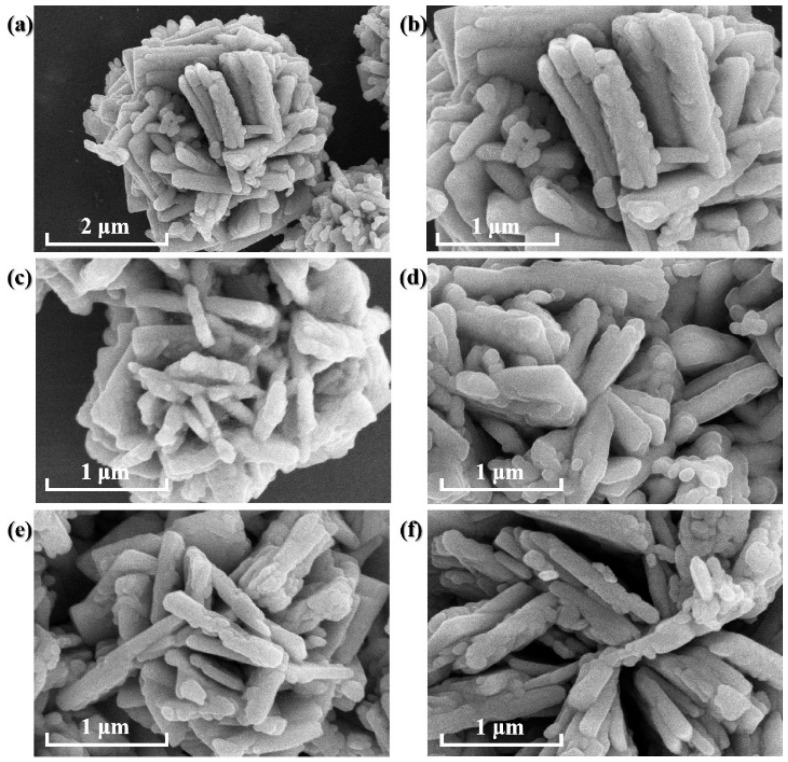
SEM images of (**a**,**b**) WO_3_; and WO_3−x_ samples: (**c**) WO_3−x_-0.5 h; (**d**) WO_3−x_-3 h; (**e**) WO_3−x_-6 h; (**f**) WO_3−x_-10 h.

**Figure 2 nanomaterials-14-00923-f002:**
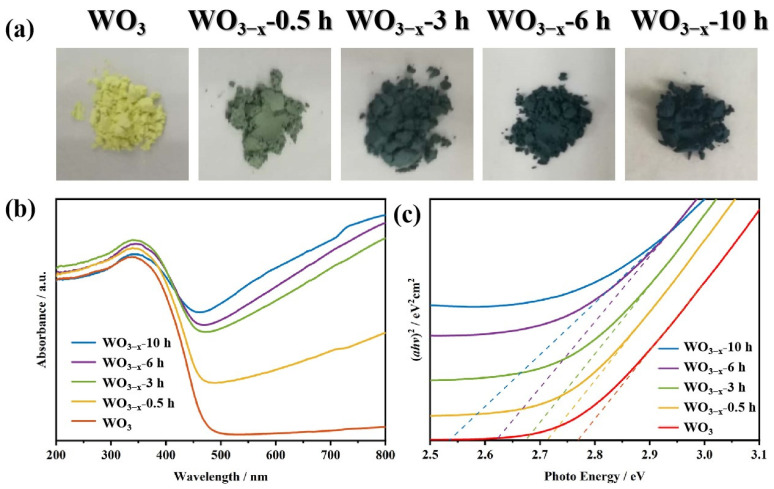
(**a**) The photographs; (**b**) UV-vis diffuse reflectance spectra; (**c**) band gaps calculated from Kubellka–Munk plots of the samples.

**Figure 3 nanomaterials-14-00923-f003:**
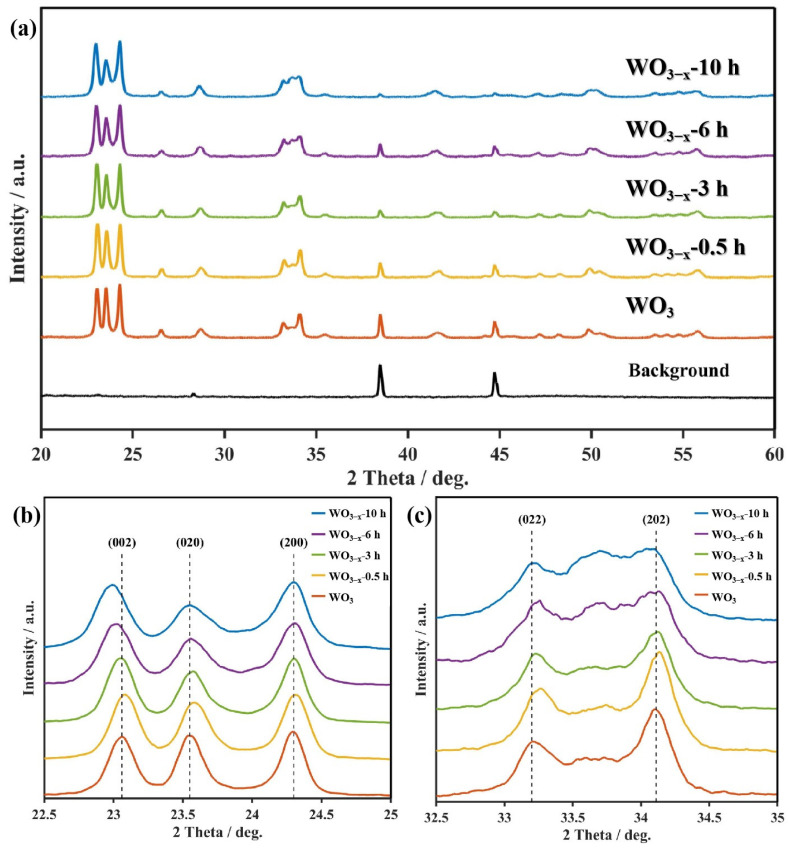
(**a**) XRD patterns of WO_3_, WO_3−x_-0.5 h, WO_3−x_-3 h, WO_3−x_-6 h and WO_3−x_-10 h; (**b**) their XRD patterns on (002), (020) and (200) planes and (**c**) on (020) and (200) planes.

**Figure 4 nanomaterials-14-00923-f004:**
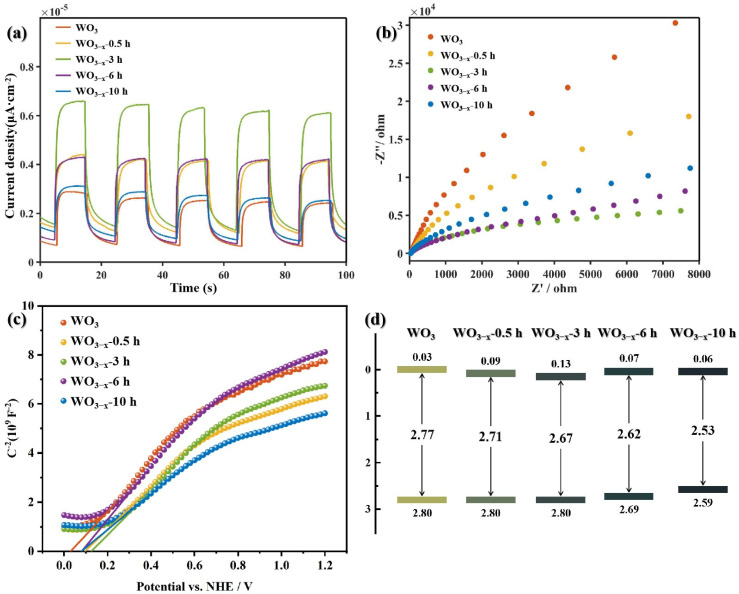
(**a**) Transient photocurrent responses; (**b**) EIS spectra; (**c**) Mott–Schottky plots; and (**d**) band structure diagram of as-prepared samples.

**Figure 5 nanomaterials-14-00923-f005:**
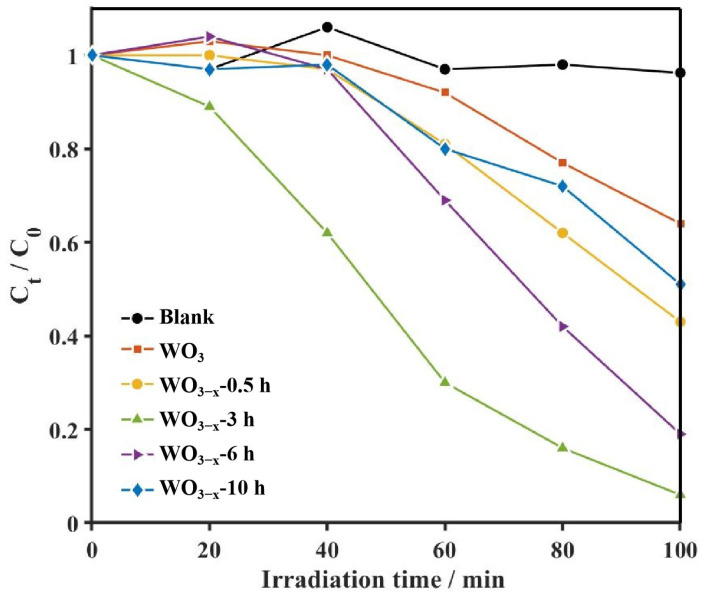
Photocatalytic degradation curve of as-prepared samples for CBZ under UV-vis.

**Figure 6 nanomaterials-14-00923-f006:**
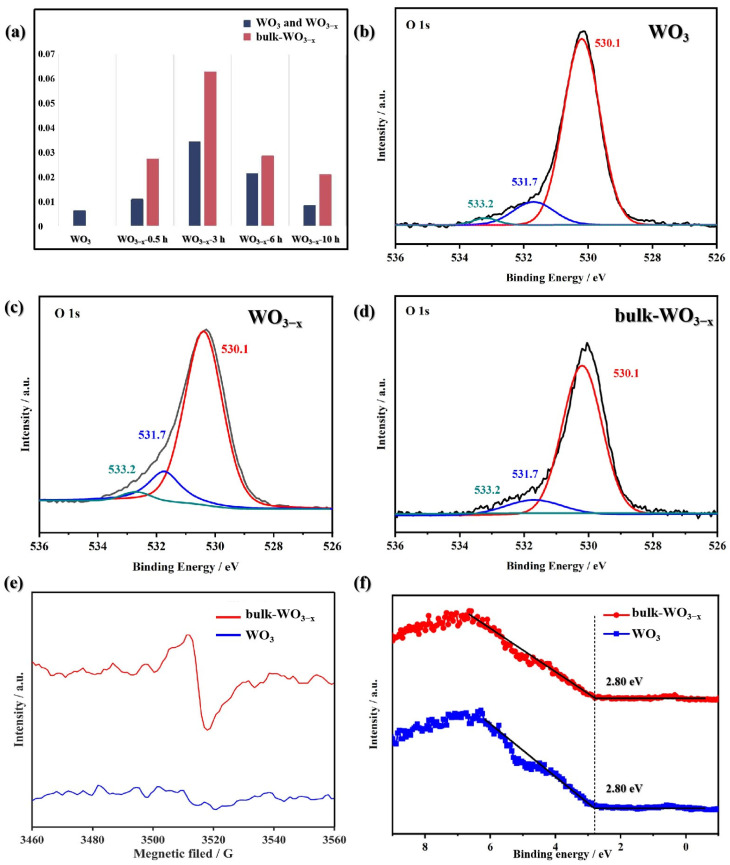
(**a**) Kinetic constant (k) of as-prepared samples for CBZ under UV-vis; O 1s XPS spectra of (**b**) WO_3_, (**c**) WO_3−x_ and (**d**) bulk-WO_3−x_; (**e**) EPR spectra and (**f**) VB XPS spectra of WO_3_ and bulk-WO_3−x_.

**Figure 7 nanomaterials-14-00923-f007:**
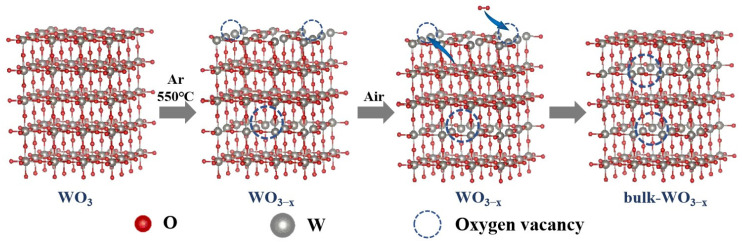
The schema of preparation from pristine WO_3_ to WO_3−x_ and bulk-WO_3−x_.

**Figure 8 nanomaterials-14-00923-f008:**
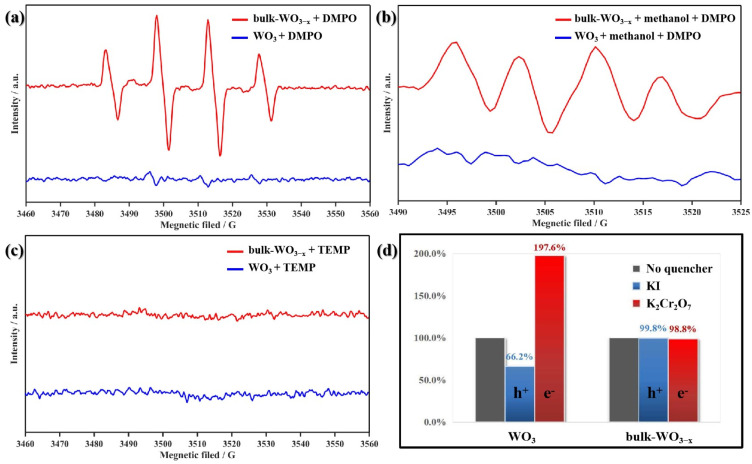
EPR spectra of (**a**) DMPO-·OH adduct; (**b**) DMPO-·O^2−^ adduct and (**c**) TEMP-^1^O_2_ for WO_3_ and bulk-WO_3−x_ under UV-vis irradiation; (**d**) the changes in degradation efficiencies of CBZ in WO_3_ and bulk-WO_3−x_ photocatalytic systems with KI or K_2_Cr_2_O_7_ as quenchers.

**Figure 9 nanomaterials-14-00923-f009:**
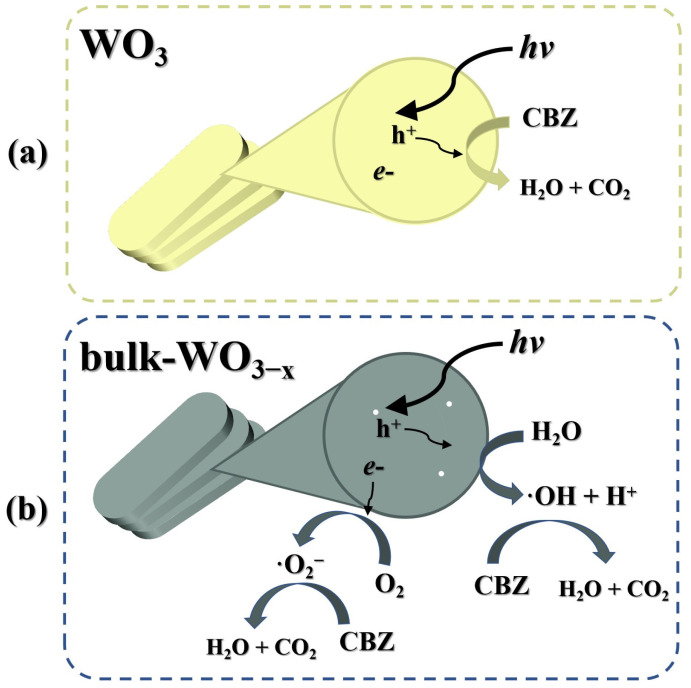
Illustration of photocatalytic mechanism of (**a**) WO_3_ and (**b**) bulk-WO_3−x_.

**Table 1 nanomaterials-14-00923-t001:** Detailed results of O 1s XPS spectra of pristine WO_3_, OV-WO_3_-3 h and bulk-OV-WO_3_.

		Total	O_L_	O_OH_	O_H2O_
Area	WO_3_	84,376.662	72,615.110	10,022.600	1738.952
	WO_3−x_	174,771.533	142,919.300	26,817.080	5035.153
	bulk-WO_3−x_	78,938.520	64,014.520	8907.012	6016.988
Percentage	WO_3_	100%	86.1%	11.8%	2.1%
	WO_3−x_	100%	81.8%	15.3%	2.9%
	bulk-WO_3−x_	100%	81.1%	11.3%	7.6%

## Data Availability

The data presented in this study are available on request from the corresponding authors.

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
