# Peer review of "A Bulk Oxygen Vacancy Dominating WO3−x Photocatalyst for Carbamazepine Degradation"

_nanomaterials, 2024, doi:10.3390/nano14110923_

Round 1

Reviewer 1 Report

Comments and Suggestions for Authors

The author rises the very good work on “A Bulk Oxygen Vacancy dominating WO3 Photocatalyst for 2 Carbamazepine Degradation” This work is very interesting, and it is also clearly presented. The subject addressed is this article is in the scope of the journal. This paper is suitable to be considered for the publication after the several following issues are addressed. I recommend the revision of this work.

Comment 1: More explanation is needed in the introduction part for where there is a research gap and what the goals of the research are.

Comment 2: Please underscore the scientific value added of your paper in your abstract and introduction.

Comment 3 Please update some very important/landmark references according to the metal elements on environment science into introduction part.

EcoMat, https://doi.org/10.1002/eom2.12044;

Chemical Engineering Journal, https://doi.org/10.1016/j.cej.2021.134104;

Comment 4: I would suggest the author to enhance the last paragraph of introduction by explaining your work performed in this article.

Comment 5: Please explain your results into steps and links to your proposed method, for example, the vacancy dominating, by introducing from deeper understanding aspects. Ref: Nano-Micro Letters, 2023, https://link.springer.com/article/10.1007/s40820-023-01152-z.

Comment 6: Please make sure your conclusions' section underscore the scientific value added of your paper.

Comment 7: The English writing of this manuscript should be intensively improved, the help from senior researcher/scientist is recommended to further improve it. There are still some minor mistakes in the manuscript.

Comment 8: Take care that "spaces are required between characters".

Comments on the Quality of English Language

English should be further improved.

Reviewer 2 Report

Comments and Suggestions for Authors

The research article: A Bulk Oxygen Vacancy dominating WO3 Photocatalyst for Carbamazepine Degradation

This research is focused on tungsten trioxide nanostructure formation with oxygen vacancies. The formed structures were tested as photocatalysts for Carbamazepine Degradation under ultraviolet-visible irradiation from a 350 W Xenon lamp. The topic of this study is novel and interesting; however, there are some remarks on how to improve the manuscript:

1.     The tungsten trioxide with internal oxygen vacancies should not be described by a general formula of WO3. If possible, it would be more adequate to use WO3-x  or an accurate structure in the whole manuscript.

2.     In the introduction, it would be worth mentioning that oxygen vacancies rich WO3 can be prepared not only by thermal treatment in an inert or reducing atmosphere but also oxygen vacancies rich WO3/WO2.9 heterostructures can be formed by sol-gel methods. See: Materials 2020, 13(3), 523; https://doi.org/10.3390/ma13030523

3.     As the research is focused on oxygen vacancies rich WO3 structures, in the introduction part, it would be worth mentioning methods such as photoluminescence: Materials 2020, 13(12), 2814; https://doi.org/10.3390/ma13122814; EPR https://doi.org/10.1016/j.electacta.2018.04.109, Raman spectroscopy DOI: 10.1039/C8CY00994E (Paper) Catal. Sci. Technol., 2018, 8, 4399-4406 frequently used and highly sensitive to the vacancies formation of oxygen vacancies in WO3-x structures.

4.     All the images should be enlarged. They are too small to read.

5.     In the MM section: “Pristine WO3 sample is prepared by a hydrothermal method as reported before” – lacks reference to the synthesis procedure.

6.     Figure 5, the control measurement of Carbamazepine under illumination should be added to evaluate the efficiency of the photocatalyst.

7.     Moreover, how was the adsorption impact on the photocatalysis measurements evaluated? Might it be that only the adsorption of the Carbamazepine on WO3-x was measured instead of photocatalysis?

8.     The photocatalysis results should be recalculated in the form of “mg dye decomposed per 1mg of photocatalyst".

9.     The photocatalysis results should be compared with other studies' results in the prepared table to show the influence of WO3-x prepared in this study compared to other studies.

Comments on the Quality of English Language

English should be double-checked.

Reviewer 3 Report

Comments and Suggestions for Authors

Comments on the manuscript “A Bulk Oxygen Vacancy dominating WO3 Photocatalyst for Carbamazepine Degradation by, W. Guo, Q. Wei, G. Li, F. Wei, Z. Hu, submitted to MDPI nanomaterials.

 The authors describe a procedure of inducing oxygen vacancies in the crystal structure of WO3 and investigate the catalytic activity of so obtained materials for the carbamazepine degradation reaction.  An optimal content the oxygen vacancies turns out to be reached by subsequent oxygen treatment.  The authors explain this as a result of surface recovery and conclude that the bulk vacancies are the most effective in enhancing the catalytic performance of the vacancy-containing WO3.  This seems to me a reasonable conclusion and it is also well supported by experimental evidence obtained using various techniques, such as EPR, XPS and others.  The article is publishable, but certain minor issues might be considered.

1)  What is the background line in the XRD patterns shown in Figure 2.

2)  Equation (1): please define the terms clearly.  I presume, that c0 is an initial concentration and ct is the one after time period t, but if so, the ratio in the brackets should be ct/c0, rather than the reciprocal of this.

3)  p. 8, line 232: “However, no drift is observed between WO3 and bulk-OV-WO3 samples, [...]”.  Would the author please explain what does “drift” mean in this sentence?

Comments on the Quality of English Language

The English of the article is very good, but there are some rather minor issues:

 p. 5, line 162: “Nevertheless, the photocurrent density of OV-WO3 samples decreases when the oxygen vacancies exceed the optimal proportion, indicating that the  presence of more oxygen vacancies in WO3 nanosheets does not necessarily further increase.  Further increase what?  Apparently something is mission in this sentence. 

Similarly in p. 3, line 66: “Importantly, when the surface oxygen vacancies are recovered, the sample with mainly bulk oxygen vacancies exhibit much higher than sample with both surface and bulk oxygen vacancies.  Do the authors mean “much higher activity”?

The are some minor issues, for instance, p. 6, line 207: “Interestingly, it was found we discovered [...]  please rephrase; either “it was found”, or “we discovered”.

Round 2

Reviewer 2 Report

Comments and Suggestions for Authors

Dear Authors,

Thank you for the improvements. I suggest the manuscript for the publishing process.